# Simultaneous synthesis of FLAIR and segmentation of white matter hypointensities from T1 MRIs

**Mauricio Orbes-Arteaga**
Translational Imaging Group
University College London
Biomediq A/S
Copenhagen Ø, Denmark
henry.arteaga.17@ucl.ac.uk

**M. Jorge Cardoso**
Translational Imaging Group
University College London
m.jorge.cardoso@ucl.ac.uk

**Lauge Sørensen**
Biomediq A/S
Department of Computer Science
University of Copenhagen,
lauges@di.ku.dk

**Marc Modat**
Translational Imaging Group
University College London
m.modat@ucl.ac.uk

**Sébastien Ourselin**
Translational Imaging Group
University College London
s.ourselin@ucl.ac.uk

**Mads Nielsen**
Biomediq A/S
Department of Computer Science
University of Copenhagen,
madsn@di.ku.dk

**Akshay Pai**
Biomediq A/S
Department of Computer Science
University of Copenhagen,
akshay@di.ku.dk

## Abstract

Segmenting vascular pathologies such as white matter lesions in Brain magnetic resonance images (MRIs) require acquisition of multiple sequences such as T1-weighted (T1-w) –on which lesions appear hypointense– and fluid attenuated inversion recovery (FLAIR) sequence –where lesions appear hyperintense–. However, most of the existing retrospective datasets do not consist of FLAIR sequences. Existing missing modality imputation methods separate the process of imputation, and the process of segmentation. In this paper, we propose a method to link both modality imputation and segmentation using convolutional neural networks. We show that by jointly optimizing the imputation network and the segmentation network, the method not only produces more realistic synthetic FLAIR images from T1-w images, but also improves the segmentation of WMH from T1-w images only.

## 1 Introduction

Segmenting white matter hyperintensties/hypointensities (WMH) from brain magnetic resonance images (MRI) have a profound impact in understand the role of vascular pathology in various

1st Conference on Medical Imaging with Deep Learning (MIDL 2018), Amsterdam, The Netherlands.

neurological disorders Caligiuri et al. [2015]. Segmenting WMH manually is not feasible due to time, and inter/intra-rater variability. In the recent years, deep learning strategies have gained attention in medical image analysis. Specially, convolutional Neural Networks(CNN) have been widely used in disease classification, segmentation, and registration tasks Litjens et al. [2017], Kamnitsas et al. [2017], Havaei et al. [2016]. Particularly, CNNs have also become the first choice in the segmentation of WMH. For example, the top-three performing methods in the WMH segmentation challenge [1], have relied on some form of CNNs.

From a more practical perspective, segmenting vascular pathologies such as WMH usually requires multiple MRI modalities Griffanti et al. [2016], Dadar et al. [2017]. Most often, in addition to the usual T1 scans, FLAIR sequences are also obtained since additional MRI sequences are specifically designed to provide complementary information to T1 scans. Having mentioned this, it is notable that most of the existing datasets only contain T1 scans or T1/T2/PD scans due to logistical reasons. Given the presence of limited data with desired multiple modalities, data imputation methods are used to learn the synthesis of missing modality using T1 scans. The intention of imputing data is to guide the optimization using prior information, i.e., the available FLAIR sequence. As stated in van Tulder and de Bruijne [2015], synthetic data helps the segmentation because of two reasons. Firstly, the flexibility of synthesis model allows finding features that can not be seen by the classifier in an otherwise single-modality model. Secondly, the size of the training set is synthetically increased which is useful in the training process.

Among CNN-based imputation methods, the most popular ones using a flavor of generative adversarial networks (GANs) Goodfellow et al. [2014]. For instance, Nie et al. [2017] use GANs to generate CT images from MRI images. However, most of the current implementations treat synthesis as a preprocessing step Ben-Cohen et al. [2018], Zhang et al. [2018], Huo et al. [2017]. This restricts the network, and the features may not be particularly useful for the final segmentation.

In this paper, we proposed a simultaneous training based synthesis method that combines generation of the missing modality and segmentation – inspired from Tran et al. [2017]. Experiments on the WMH segmentation challenge 2017 dataset shows that using the proposed method to synthesize FLAIR images, we not only obtain higher quality synthetic flair images (when compared to treating synthesis a preprocessing step) but also improve the segmentation of WMH using T1-w images only.

## 2 Methods

Let $\mathcal{X}=\{\boldsymbol{X^n}, \boldsymbol{L^n} : 1, \ldots, N\}$ be an annotated training set which have $N$ subjects . Here, $\boldsymbol{X} = \{X_a, X_b\}$, is a pair of MRI images from two different modality sources for a given subject, and $\boldsymbol{L}$ is a volume with the manual annotation for WMH. The goal in multi modal segmentation task is to find a mapping $C(\boldsymbol{X}, \theta_c)$ from a pair of available modalities to a corresponding segmentation.

$$C : \{X_a, X_b\}, \to L \tag{1}$$

Here, $C$ is a function represented by a CNN with parameters $\theta_c$. We then train $C$ to maximize:

$$\max_{\theta_c} \mathbb{E}[\log\ p(L|X_a, X_b, \theta_c)] \tag{2}$$

It is evident that to train, and subsequently test such a scheme, both modalities are needed. This is a restriction, specially when the network is used to test retrospective data with missing modalities. One of most common approaches to deal with missing modalities is to impute them. Formally, a function $G$ (a CNN) is trained to learn a mapping between the available modality and the missing modality., i.e $G(X_a) \approx X_b$. Subsequently, the synthesized modality is used in conjunction with the available modality to train a classifier for segmentation. The optimization function for the classifier in Equation (2) can be re-written as:

$$\max_{\theta_c} \mathbb{E}[\log\ p(L|X_a, G(X_a), \theta_c)] \tag{3}$$

Note that in this scheme, the generation and the classification are different optimizations. No complementary information is taken into account. Therefore, in this work, we aim to learn the

---

[1] http://wmh.isi.uu.nl

generation and classification (respectively performed by $G$ and $C$) simultaneously so that $C$ reinforces the generation $G$ to produce not only realistic images but also relevant features that help in the optimization of $C$.

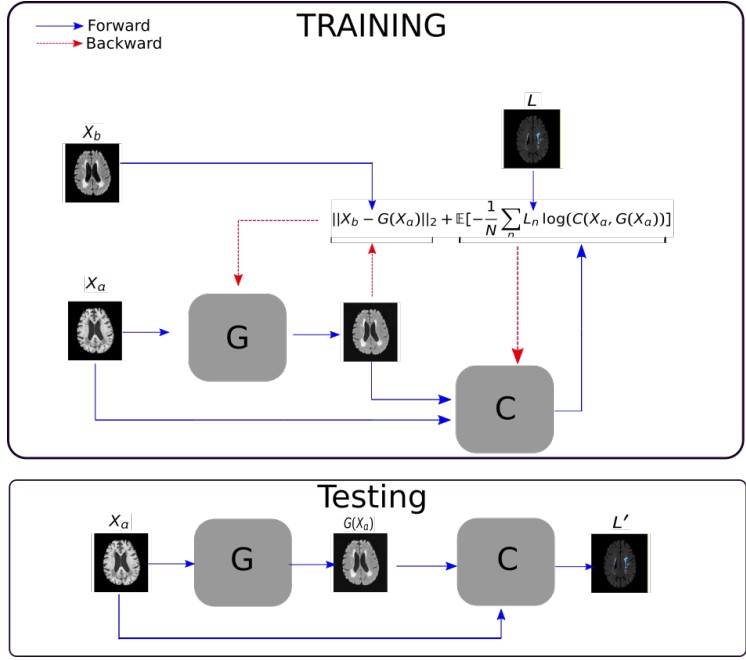

Figure 1: Illustration of process follow for training and testing of our method.

The scheme is basically composed of two networks, a generator $G$ and a classifier $C$ where both networks are trained end to end iteratively, see Figure 1. The classifier training is linked to the generator by taking both the real T1 image denoted by $X_a$ and the generated image $G(X_b)$ to produce a segmentation $\boldsymbol{L'}$. The loss function of the classifier network is:

$$\mathcal{L}_C = \mathbb{E}[-\frac{1}{N} \sum_n L_n \log(C(X_a^n, G(X_a^n)))] \tag{4}$$

In order to train $G$ to produce images looks like as FLAIR, we use L2 as a reconstruction error between the real missing modality image and its corresponding generation. One may then view the classifier to be a regularization term to the generator or vice versa. The L2 loss for the generator is given by:

$$\mathcal{L}_G = ||X_b - G(X_a)||^2 + \mathbb{E}[-\frac{1}{N} \sum_n L_n \log(C(X_a^n, G(X_a^n)))] \tag{5}$$

## 2.1 Network architectures

We use U-Nets Li et al. [2018] (winner in 2017 MICCAI- WMHs segmentation challenge) as the segmentation network, and a modification of it as a generation network. The changes involve changing the number of inputs channels from two to one which corresponds to the T1 modality, we also change the `Sigmoid` function in the final layer by `LeakyRelu`. We use Adam optimizer with learning rate `0.0002` for both the networks, and batch normalization. The classifier and generator are trained iteratively with the same frequency. We do not use any data augmentation.

# 3 Experiments and results

## 3.1 Data and Experiment

We validated our proposed method on the training dataset from the 2017 White Matter Hyperintensity Segmentation Challenge (`http://wmh.isi.uu.nl`). This dataset is composed of T1 and FLAIR scans for 60 subjects from three different clinics (Utrecht, Singapore, and AmsterdamGE3T, 20 subjects for each one), the data is complemented with manual annotations of WMH from presumed vascular origin. FLAIR images have been used as a reference for label annotations, so, T1 images have been registered to this space. The images were also corrected for bias field inhomogeneities using SPM12. As a further preprocessing we use only two of three stages performed in Li et al. [2018], which include *i)* cropping or padding of axial slices *ii)* Gaussian normalization of voxel intensities.We did not perform data augmentation as these did not show significant improvement in segmentation.

All the methods were evaluated using a 6-fold cross validation. The dataset was split in such a way that all the 60 images are tested at least once. For each fold, we pick 10 subjects for test, 5 for validation, and the remaining 45 are used for training. For evaluation, dice scores (DSC), false positive rates (FPR), and false negative rates (FNR) are used.

## 3.2 Results

We evaluated our method in segmenting WMH from T1-w images using: a) Synthesized FLAIR images by treating the synthesis as a preprocessing step – we will refer to this method as *offline synthesis*; b) Synthesized FLAIR images using the proposed method, and c) without any synthesis – we will refer to this method as *Unimodal*. Baseline methods are illustrated in Figure 2

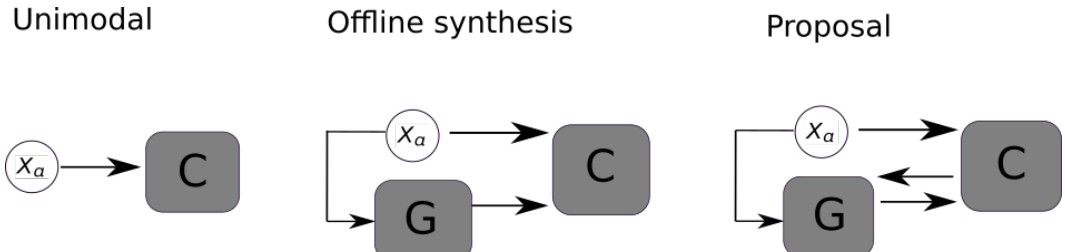

Figure 2: Illustration of methods of comparison, $X_a$ represent a T1 image.

Table 1 shows the mean of each measure for all considered methods. As we can see, our method achieves higher dice scores than baseline methods. A mean dice improvement of nearly three percent is obtained using our proposed method when compared the baseline method without any imputation. In addition, the proposed method also improves segmentation when compared to an offline synthesis.

Table 1: Average of performance measures for all comparison methods, results in bold are significantly different (p<0.005) from the baseline *Unimodal* method (top row)

|  | Evaluation Metric | | |
| Method | DSC(%) | FPR(%) | FNR(%) |
| --- | --- | --- | --- |
| Unimodal | 55.99 | 78.67 | 38.06 |
| Offline synthesis | 54.42 | 63.50 | **43.39** |
| Proposal | **57.81** | **58.20** | **41.33** |

It is important to note, that our proposed method shows a FPR $20.47\%$ lower than *Unimodal* and $5.3\%$ lower than *offline synthesis* method, showing the effectiveness of our method to reduce the number of false positives. On the other hand, *Unimodal* method shows the lower rates in terms of FN.

In order to better understand the above results, we visually analyzed the output segmentation performed for each method. Table 2 shows the results for three different slices (one slice per column).

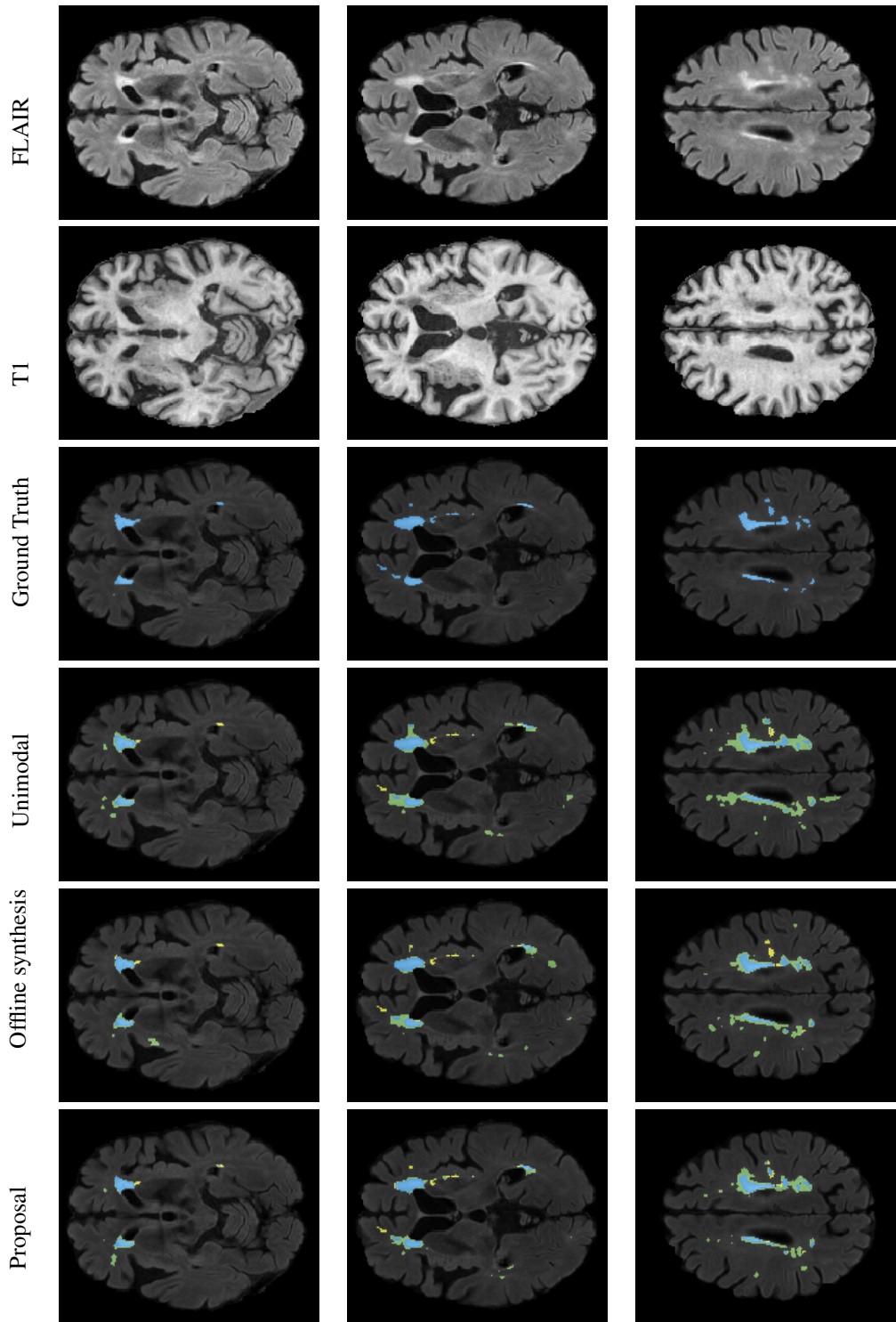

Table 2: Segmentation results for all proposed methods, each column represent a different slide in the image, blue areas are regions which were correctly labeled, false positives are shown in green, and false negatives in yellow

As illustrated, the proposed method is able to produce less false positives. It is also important to note that, unimodal segmentation is the one that produces more false positives, showing the advantage of

using synthetic data. Regarding the nature of false positives, it can be easy to see in the third column a large number of false positives are on the border of periventricular lesions for the *Unimodal* method in comparison to the proposed method. Also from the first and second column, it can be observed that *Unimodal* tend to produce more small regions of false positives near to cortical areas. Removing such false positives requires additional post-processing steps, therefore, it is of value avoid this kind of over-segmentation. It can also be noted that synthesis methods tend to produce the same kind of false negatives, this may be due to the blurring effects in synthesized images since the information available during testing is limited – which otherwise is available from a FLAIR sequence.

### 3.3 Results of Generation

Here we compare the generate FLAIR images obtained for the generator using our optimization strategy against the generated images obtained for using *off-line synthesis*. Firstly, images were quantitatively evaluated in terms of reconstruction using two well known measures, namely mean absolute error(MAE) and peak-signal-to-noise-ratio(PSNR). Results of reconstruction measures are shown in Table 3 , as we can see our proposal outperforms the baseline approach in both MAE and PSNR. Specifically, images generated for our proposed method achieve an average PSNR of 11.01 which is considerably higher compared with 9.65 obtained for images generated *Offline*. Reconstruction superiority of our methods is confirmed by the MEA results, 0.26 and 0.31 for our proposal and the baseline respectively.

Table 3: Average MAE and PSNR between real Flair images and the synthetic images generated for each method

| | Method | |
|---|---|---|
| Measure | Offline synthesis | proposal |
| MAE | 0.3153 | 0.2566 |
| PSNR(DB) | 9.65 | 11.01 |

In order to analyze qualitatively the results of our generator, we extract slices with different WMHs loads, Table 4 shows the reconstruction results for three different levels of loads. As we can see in the first row, both methods produce a similar response in regions with a low load of lesions, it can be observed that generated images are similar to the real FLAIR images in the left, and these not present evident structural distortions. However, it can be noted images exhibit blurred effects, which can be due to L2 based optimization, more complex generative networks with adversarial loss optimization as GANs tend to eliminate blurred effect but at the expense to produce structural distortions. In the application presented in this work it is important to preserve the structural information, thus, our L2 based optimization present a good balance between preserve structural information and blurred effects. In the second and third column, it can be observed the performance of both methods when facing the presence of lesions, as can be seen, both methods have a good response to large and contiguous lesions. It also can be noted both methods tend to produce poor performance in small and diffuse WMHs marked in red, note, these lesion do not exhibit identifiable patterns in T1 images, however it can be seen that our proposed method is more sensitive to these patterns which enable to highlight some small regions as those marked in green.

## 4   Discussion and Concluding Remarks

In this paper, a new CNN-based method to improve WMH segmentation from T1-w images alone is proposed. The method jointly performs imputation and segmentation in such a way that both tasks are mutually benefited. To this end, FLAIR sequences are used to drive the optimization, which reflects in the results where joint optimization of synthesis and segmentation yield better segmentation from T1-only images.

From segmentation results in Section 3.2, it is evident that the T1-based segmentation tends to have excessive over-segmentation of images. By using prior information from FLAIR images through a generator, we are able to reduce the number of false positives. However, it could be observed that, if imputation comes from an independent synthesis model, images tend to be under segmented (high

| FLAIR | T1 | Offline synthesis | Proposal |
|-------|-----|-------------------|----------|

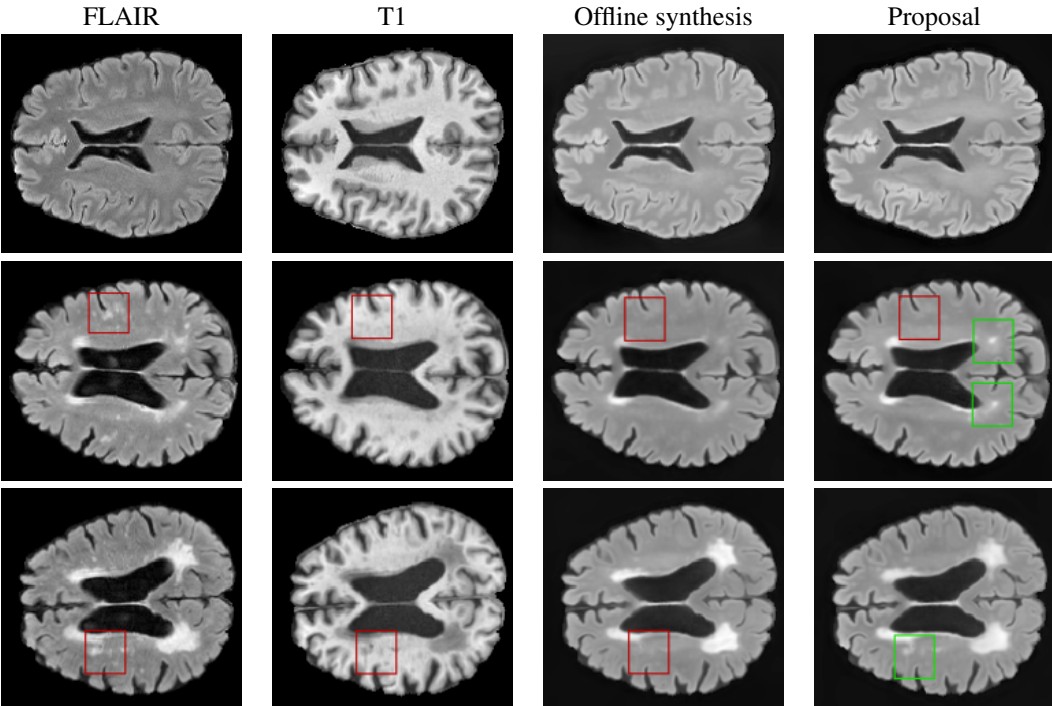

Table 4: Results of Generation for all the proposed methods,

FNR) reducing the overall segmentation accuracy. The proposed joint optimization strategy better adapts to capture small lesions, which leads to significantly better overall segmentation performance.

In addition to an improved segmentation performance, we can see in Section 3.3 that the proposed method also produces better synthetic FLAIR images when compared to networks that trained to only specialize in generation. This may be due to the complementary information available through a joint optimization with the segmentation network. Specially, lesions that are barely visible in T1 images are seen in synthetic images produced by the proposed method.

One of the disadvantages of our method is using L2 as a loss function can produce blurring effect on the images. Using adversarial training by the use of a discriminative network as a loss function may overcome this issue. However, with an introduction of an additional network and the availability of limited training data, the optimization may be prone overfitting. Therefore the proposed method with L2 loss provides a good compromise between the complexity of the model and segmentation performance.

## Acknowledgments

This project has received funding from the European Union's Horizon 2020 research and innovation programme under the Marie Skłodowska-Curie grant agreement No 721820. We would like to thank both Microsoft and NVIDIA for providing computational resources on the Azure platform for this project.

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
