# OpenReview forum: "Simultaneous synthesis of FLAIR and segmentation of white matter hypointensities from T1 MRIs"
_MIDL.amsterdam/2018/Conference — MIDL 2018 Poster_

### Review · AnonReviewer3 · 2018-05-08
**One of the most convincing way to integrate synthesis in image analysis**

**Rating:** 4
**Confidence:** 3

**Review:**

Authors propose joint image synthesis and segmentation to improve segmentation performance using one modality in tasks where multiple modalities are often used. The work is motivated by imputation by synthesis literature, which received attention from the research community in the recent years.

Pros:
1. The method makes a lot of sense. Authors include synthesis almost as a regularisation term in the optimisation. I really like this approach.
2. Evaluation is proper. Authors compare the proposed method with an appropriate baseline as well as the obvious alternative, performing synthesis as a pre-processing step before segmentation.
3. Results show modest but convincing improvement.
4. Article is well written.

Cons:
1. In this application, the main task is segmentation. Synthesis’ role is to improve segmentation. Therefore, the value of synthesis itself is unclear. Authors could have motivated synthesis itself more to justify the last analysis in the experimental section.
2. Literature review in synthesis is missing important work both from pre-deep learning era, using patch-based and forests, and from deep learning era, which is simply sad because there is no limit in references in these articles and giving credit to people who contributed to this line of work seems to be the proper thing to do. If there is space for changes, I really suggest a proper literature review giving credit to researchers who pushed synthesis in medical image analysis.

**Special Issue:**

Yes

---

### Review · AnonReviewer1 · 2018-05-09
**The authors propose to use CNN for joint imputation of missing FLAIR segmentation of white matter hyper-intensities (WMH) based on corresponding T1-weighted MR images. The proposed method was evaluated on the WMH segmentation challenge 2017 dataset with very limited comparison with the state-of-the-art methods.**

**Rating:** 1
**Confidence:** 3

**Review:**

Overall, the technical contribution of this paper is limited. In addition, presented experimental results are insufficient to support the conclusion. My main concerns are listed below.

1)	The authors aim to impute an additional modality (i.e., FLAIR), but the author did not provide comparison with using both the real FLAIR and T1 for segmentation.

2)	The architecture of G and C should be more clearly explained. It seems that the loss function in Eq. (3) and (4) is incorrect. What’s the relationship between X_a and X_a^n?

3)	It would be better to more clearly indicate the contribution and novelty of this paper. To my understanding, the major contribution of this paper is the imputation of the missing FLAIR from T1, while there is no motivation behind since the authors focus on the dataset with real FLAIR images.

4)	What’s the size(s) of the images? Please give more details about the dataset and the preprocessing step.

5)	To demonstrate the performance improvement due to generated FLAIR images, it is important to compare the performance with that obtained by using only real FLAIR images.

6)	I strongly suggest the authors to compare the proposed method with the state-of-the-art methods to show its effectiveness and contribution.

7)	I also suggest the authors to report the results of each fold.

8)	The manuscript contains a lot of typos.


**Special Issue:**

No

---

### Review · AnonReviewer2 · 2018-05-09
**Simultaneous synthesis of FLAIR and segmentation of white matter hypointensities from T1 MRIs**

**Rating:** 4
**Confidence:** 3

**Review:**

This paper presents an approach to segment white matter lesions in MRI images.
At the same time, it tackles the problem of generating missing MRI image modality, namely FLAIR.
The core of the contribution is a method to simultaneously producing a segmentation map and generating a missing modality image, based on a GAN approach.
Compared to a unimodal approach (only using available modality) and to an offline synthesis of missing modality, the proposed approach achieves higher dice score.

The core idea of the paper is very interesting, and the proposed approach is effective.
However, when compared to the state of the art on the same dataset, the reported results in terms of Dice score are much lower (57% vs 80%).
It would be interesting if the authors could comment on how to further develop this approach in order to improve the segmentation performance.

Authors should check the text for presence of repeated words and for the formatting of references.


**Special Issue:**

No

---

### Decision · Program_Chairs · 2018-05-15
**Paper104 Acceptance Decision**

Poster